# Melatonin-Mediated Regulation of Antioxidant Defense Enhances the Resistance of Tea Plants (*Camellia sinensis* L.) to Lead-Induced Stress

**DOI:** 10.3390/plants14101417

**Published:** 2025-05-09

**Authors:** Jianwu Li, Jiao Yang, Xin Liang, Shuping Zhan, Yixuan Bai, Li Ruan

**Affiliations:** 1College of Environment and Resources, Carbon Neutrality, Zhejiang A&F University, Hangzhou 311300, China; 2Institute of Sericulture and Tea, Zhejiang Academy of Agricultural Sciences, Hangzhou 310021, China

**Keywords:** tea plant, lead stress, melatonin, antioxidant capacity, anthocyanin

## Abstract

Lead (Pb) is a toxic heavy metal that severely impairs plant growth and crop quality. Melatonin, a widely present indoleamine compound, enhances plant stress tolerance, yet its role in tea plant resistance to lead stress remains unclear. This study examined two tea genotypes with distinct anthocyanin levels, Longjing43 (LJ43) and Zijuan (ZJ), comparing their phenotype, antioxidant capacity, secondary metabolite synthesis, and lead transport gene expression under lead stress. Excessive Pb exposure caused severe oxidative stress, reducing PSII efficiency, increasing ROS accumulation, and intensifying lipid peroxidation. ZJ, with higher anthocyanin concentration, exhibited stronger lead stress resistance than LJ43. Under lead stress, melatonin promoted phenylalanine accumulation in ZJ, facilitating its conversion into anthocyanins and catechins via key gene regulation (*CsC4H*, *CsLAR*, and *CsANS*). Moreover, exogenous melatonin significantly reduced Pb concentrations in roots, stems, and leaves, with a more pronounced effect in ZJ (reductions of 20.46%, 53.30%, and 38.17%, respectively), which might be associated with the downregulation of Pb transport genes like *CsZIP1* (notably showing a 29-fold decrease). While these results suggest that melatonin might enhance Pb stress tolerance by modulating flavonoid metabolism and restricting Pb uptake, the specific roles of anthocyanins and catechins in this process remains to be fully elucidated. Further studies are necessary to clarify the primary bioactive compounds and mechanisms involved in melatonin-mediated heavy metal stress mitigation.

## 1. Introduction

Pb is a highly toxic heavy metal with high damaging properties to the nervous and reproductive systems. Pb primarily originates from sources such as automobile exhausts, petroleum combustion, mining, smelting, and the use of Pb-containing fertilizers and pesticides. It is widely distributed in the atmosphere, water, and soil, posing significant threats to ecosystems and human health through the food chain [1,2]. Although Pb in food, vegetables, and fruits is well recognized as harmful [2,3,4,5,6,7], there is little focus on beverage crops and relevant mitigation strategies. In terms of global consumption, tea is the second most popular non-alcoholic beverage after water. Despite the ability of tea plants to grow in a wide range of environmental conditions, they are still highly susceptible to environmental stress, including drought, low temperatures, and heavy metals. These stresses impair their physiological functions and metabolic activities [8]. These adversities affect the yield, quality, and safety of tea plants, which is an important limiting factor for the popularization of tea.

Recent studies indicate that soil Pb concentrations in several countries have already surpassed permissible limits [9]. Moreover, soil acidification increases the availability of soluble and exchangeable Pb, thereby intensifying Pb uptake by tea plants [10]. Pb stress not only alters the ultrastructure of cellular organelles but also induces physiological and biochemical disruptions in plants [11]. Excessive Pb accumulation is highly toxic to plants, impairing photosynthetic efficiency, inhibiting growth, and ultimately reducing yields [5,12]. At the cellular level, Pb exposure leads to excessive ROS production, resulting in oxidative damage and further compromising plant health [13,14]. These damages can reduce the yield and quality of tea. Pb contamination has been detected in commercial tea products from China and numerous other countries [15]. During brewing, Pb can leach into the tea infusion, posing a potential risk to human health [16,17]. This underscores the urgent need to develop effective strategies to minimize Pb accumulation in tea plants and ensure food safety.

When plants are subjected to Pb stress, they employ various defense mechanisms to counter the external threat. The first line of defense is physical barriers such as thick cuticles, trichomes, and cell walls, which prevent Pb ions from entering the cells [18]. Once Pb ions enter plant tissues and cells, plants activate multiple defense mechanisms. One important mechanism for metal detoxification and sequestration is metal chelation. Plant-produced chelators, such as phytochelatins (PCs) and organic acids (e.g., citric acid, oxalic acid), bind to Pb ions to form stable complexes or transport excess metal ions into vacuoles for storage, reducing their upward movement in the plant [19,20]. Metallothioneins, low-molecular-weight proteins, play a key role in ROS scavenging and metal homeostasis [21]. Additionally, plants typically produce high levels of proline in response to abiotic stress, which helps with osmoregulation, recovery, and signaling [22]. Compounds secreted by plant roots also play a crucial role in metal tolerance. For instance, under aluminum stress, buckwheat roots secreted oxalic acid, which bound with aluminum to form a non-toxic complex, thus reducing the harmful effects of aluminum on the plant [23].

Melatonin is a potent antioxidant plant hormone that scavenges ROS, alleviates oxidative stress, regulates biological rhythms, and synergizes with other antioxidants to enhance plant stress tolerance [24,25]. Previous studies showed that melatonin could enhance plant tolerance to abiotic stresses such as water, salinity, and heavy metals by increasing the photosynthetic rate, promoting growth, and delaying leaf senescence [26,27,28,29,30]. Regarding heavy metal stress, previous research indicated that melatonin improved plant resistance to metal toxicity by scavenging ROS, increasing antioxidant enzyme activity, and enhancing photosynthetic efficiency [31,32,33]. Exogenous application of melatonin could alleviate cadmium (Cd) stress by improving physiological imbalance, promoting the synthesis of secondary metabolites, and activating the antioxidant defense system [34]. Moreover, recent studies demonstrated that 100 μmol L^−1^ melatonin significantly mitigated Pb toxicity. Melatonin enhanced Pb tolerance in maize by mediating endogenous nitric oxide (NO) signaling, while in Bermuda grass it reduced ROS accumulation by boosting antioxidant enzyme activity and increasing non-enzymatic antioxidant concentration [35,36]. Additionally, melatonin effectively alleviated Pb stress in naked oat [37] by inducing DNA demethylation of metal transporter and antioxidant genes, and also mitigated the toxic effects of lead on maize [38] and *Ardisia* species [39].

Melatonin can enhanced anthocyanin concentration in plants by inducing the transcription of most anthocyanin biosynthesis genes and anthocyanin-related transcription factors [40,41], A similar effect has also been observed for catechins. Li et al. found that melatonin increased the transcription levels of catechin biosynthesis genes in tea leaves under moderately high temperature stress [42]. Anthocyanins and catechins are both synthesized through the flavonoid biosynthetic pathway. Catechins are a major component of tea leaves, accounting for more than 75% of the tea polyphenols [43]. They are primarily concentrated in chloroplasts and cell walls, where they exhibit powerful antioxidant properties. Catechins act as scavengers of reactive oxygen species and metal ion chelators, and they play a role in regulating photosynthesis. Additionally, they help protect plants from pathogens and environmental stress [44,45]. Anthocyanins, which are widely distributed in various plant tissues, also function as antioxidants. They participate in both biotic and abiotic stress responses and can act as metal chelators [46].

Melatonin has been widely studied for its role in enhancing plant tolerance to various abiotic stresses, such as drought, salinity, and low temperature, and it has been shown to alleviate the toxic effects of some heavy metals (e.g., cadmium, copper) [47,48]. However, the specific roles and molecular mechanisms of melatonin in tea plants’ response to Pb stress remain insufficiently explored. Currently, the differences in response among different tea plant varieties under Pb stress, the regulatory mechanisms of melatonin in enhancing Pb tolerance, and its impact on Pb absorption, transport, and accumulation are not well understood. This study aims to explore the mechanism by which exogenous melatonin enhances Pb tolerance in tea plants. Using pharmacological approaches and two divergent tea cultivars, we found that the tolerance of tea plants to Pb largely depends on the basal levels of anthocyanin. Moreover, we found that melatonin might enhance Pb stress tolerance by modulating flavonoid metabolism and restricting Pb uptake. This study provides new theoretical and practical guidance for enhancing tea plants’ tolerance to heavy metal pollution.

## 2. Results

### 2.1. Exogenous Melatonin Alleviates Pb Stress in Tea Plants by Reducing Pb Accumulation and Enhancing Photosynthetic Performance

The phenotypic changes in LJ43 and ZJ under Pb stress after the application of melatonin were observed (Figure 1A). No significant phenotypic differences were observed between the CK and the MT before and after stress; however, the leaves of tea plants in the Pb group showed significant wilting, yellowing, and sparse leaves after 10 d Pb stress. The symptoms of Pb stress were notably alleviated in the treatment of MT+Pb, with reduced yellowing of the leaves, indicating that exogenous melatonin partially mitigated the damage to tea leaves caused by Pb stress. To further explore the internal mechanism changes in tea plants, we measured the chlorophyll fluorescence, chlorophyll concentration, and Fv/Fm change rates of mature leaves in LJ43 and ZJ on days 1, 4, 7, and 10 after treatment. The results showed that Pb stress weakened photosynthesis and reduced chlorophyll concentration in tea plants. After exogenous melatonin treatment, the chlorophyll concentration on day 10 increased by 5.50% in LJ43 and 11.68% in ZJ compared to Pb stress alone. The Fv/Fm change rates also improved, indicating that exogenous melatonin enhanced the photosynthetic capacity of both tea plant varieties under Pb stress (Figure 1B–D).

Furthermore, analysis of Pb concentration in the roots, stems, and leaves of tea plants under different treatments showed that the Pb concentration followed the following order: roots > stems > leaves. Compared to the Pb-only treatment, the MT+Pb treatment significantly reduced the Pb concentration in tea plants. In LJ43, the Pb concentration in the roots, stems, and leaves decreased by 10.91%, 33.49%, and 40.82%, respectively, while in ZJ, it decreased by 20.46%, 53.30%, and 38.17%, respectively (Figure 1E–G). These findings indicated that exogenous melatonin effectively reduced Pb accumulation in tea plants under Pb stress and improved their photosynthetic capacity.

### 2.2. Exogenous Melatonin Enhances Antioxidant Defense to Mitigate Pb-Induced Oxidative Damage in Tea Plant Cells

To investigate the mechanism by which exogenous melatonin alleviates Pb-induced oxidative stress in tea plants, we analyzed the total antioxidant capacity of the tea plants. We found that Pb stress significantly reduced the DPPH and FRAP levels. Compared to the Pb-only treatment, the combined treatment significantly enhanced the antioxidant capacity of the tea plants. In LJ43, the DPPH and FRAP values increased by 37.50% and 50.99%, respectively, while in ZJ, the DPPH and FRAP values increased by 79.24% and 76.77%, respectively (Figure 2A).

This study found that exogenous melatonin enhances the total antioxidant capacity of tea plants under Pb stress through two mechanisms: increasing the activity of antioxidant enzymes and promoting the levels of non-enzyme antioxidants. Under Pb stress, the SOD activity of the LJ43 and ZJ cultivars decreased by 47.43% and 61.57%, POD activity decreased by 77.20% and 62.74%, and CAT activity decreased by 24.60% and 51.64%, respectively. After exogenous melatonin treatment, the antioxidant enzyme activities were significantly increased. The SOD activity of LJ43 and ZJ increased by 63.26% and 30.30%, POD activity increased by 300.00% and 71.04%, and CAT activity increased by 37.30% and 57.50%, respectively. Moreover, under Pb stress, the glutathione redox ratio (GSH/GSSG) in the LJ43 and ZJ tea plants decreased by 86.56% and 84.60%, respectively, compared to the control group. However, under the combined treatment of melatonin and Pb, the GSH/GSSG ratio significantly increased, with an increase of 147.42% in LJ43 and 233.33% in ZJ. Aside from SOD and POD, exogenous melatonin improved the antioxidant capacity of ZJ under Pb stress more effectively than in LJ43 (Figure 2B–E).

Compared with CK, O_2_^−^ and H_2_O_2_ in LJ43 under Pb stress increased by 12.95% and 66.31%, respectively, while the corresponding increases in ZJ were 63.53% and 59.70%. After exogenous melatonin treatment, the H_2_O_2_ concentration in LJ43 showed no significant difference compared to Pb treatment alone, while O_2_^−^ concentration decreased by 15.96%. In ZJ, O_2_^−^ and H_2_O_2_ concentrations decreased by 20.63% and 35.21%, respectively. Exogenous melatonin significantly reduced ROS accumulation, with the alleviation effect in the ZJ cultivar being more pronounced than that in LJ43 (Figure 2F,G). To assess the oxidative damage caused by ROS to biological membranes, the MDA concentration was analyzed. The results showed that Pb stress significantly increased MDA levels, while the application of exogenous melatonin significantly reduced MDA levels in stressed tea plants, suggesting that melatonin effectively alleviates oxidative damage caused by Pb stress (Figure 2H).

### 2.3. Exogenous Melatonin Regulates Osmotic Regulators and Secondary Metabolites, Enhancing the Tolerance of Tea Plants to Pb Stress

Osmoregulation substances play a crucial role in helping plants cope with various stresses, such as drought, salt stress, and heavy metal stress, by maintaining cellular osmotic balance [49,50]. Under Pb stress, the proline concentration in both the LJ43 and ZJ tea plants increased significantly, rising by 108.10% and 39.51%, respectively, compared to the control (CK). This increase in proline is a typical early response mechanism to stress. However, following treatment with exogenous melatonin in combination with Pb stress, proline concentration in LJ43 and ZJ decreased by 36.36% and 46.23%, respectively, indicating that melatonin has a significant regulatory effect on proline levels. Under Pb stress, soluble protein concentration increased by 8.61% in LJ43, while no significant change was observed in ZJ. After melatonin application, soluble protein concentration increased by 4.12% in LJ43 and 4.61% in ZJ, compared to the Pb-only treatment.

In addition, plant secondary metabolites play a crucial role in responding to heavy metal stress [51,52]. The results of this study showed that under Pb stress, compared to the control, the concentrations of polyphenols and flavonoids in LJ43 significantly decreased by 11.49% and 27.62%, respectively. In contrast, in ZJ, polyphenol concentration increased by 5.22%, while flavonoid concentration decreased by 22.33%. Under the combined treatment of melatonin and Pb (MT+Pb), the polyphenol concentrations in LJ43 and ZJ increased by 6.51% and 6.59%, respectively. Notably, the flavonoid concentration in ZJ showed a remarkable increase of 51.95%, which was substantially higher than the 26.28% increase observed in LJ43 (Figure 3B,C).

### 2.4. Exogenous Melatonin Enhances the Anthocyanin and Catechin Synthesis and Reduces the Pb Transportation

The application of exogenous melatonin significantly increased the levels of endogenous melatonin in the tea plants, indicating that the exogenous melatonin was converted into its endogenous form (Figure 4A–C). In LJ43, the endogenous melatonin concentrations in roots, stems, and leaves under the MT+Pb treatment were 13.01, 173.53, and 36.32 times higher than those under the Pb treatment, respectively. In ZJ, the corresponding increases were 3.61, 99.65, and 185.34 times, respectively. Due to the addition of exogenous melatonin, the increase in endogenous melatonin in the leaves of ZJ was significantly greater than that in LJ43, whereas in roots and stems the increase was more pronounced in LJ43 than in ZJ. In addition, in the ZJ variety, which has a relatively high baseline level of anthocyanins, the application of exogenous melatonin significantly enhanced the anthocyanin concentration in the leaves. However, in the LJ43 variety, which has a lower baseline anthocyanin concentration, exogenous melatonin had no significant effect on leaf anthocyanin concentration (Figure 4D).

Moreover, we measured the concentrations of several key substances in the anthocyanin biosynthesis pathway in tea plants. Phenylalanine, as a precursor, undergoes a series of transformations, forming two distinct branches at the point of leucocyanidins production—one directed towards anthocyanin synthesis and the other towards catechin synthesis. The phenylalanine concentration in ZJ was 43.78% higher in the MT+Pb treatment than that in the Pb treatment, while the phenylalanine concentration in LJ43 was 82.56% lower in the MT+Pb treatment than that in the Pb treatment. Under Pb stress, exogenous application of melatonin could improve the phenylalanine concentration in ZJ, whereas this measure decreased the phenylalanine concentration in LJ43 (Figure 4E). Compared to CK, Pb treatment led to an 8.45-fold increase in the catechin concentration for LJ43, while Pb treatment led to a 23.19% decrease in the catechin concentration for ZJ. The catechin concentration in ZJ and LJ43 were 67.67% and 9.68% higher in the MT+Pb treatment than those in the Pb treatment, respectively (Figure 4F). Regarding epicatechin concentration, the LJ43 variety showed a 31.76% reduction under the MT+Pb treatment compared to the Pb treatment, whereas the ZJ variety exhibited an 83.71% increase in epicatechin concentration under the MT+Pb treatment compared to the Pb treatment (Figure 4G). In summary, under Pb stress, exogenous melatonin enhanced catechin accumulation while reducing epicatechin concentration in LJ43. In contrast, exogenous melatonin significantly promoted the accumulation of both catechin and epicatechin in ZJ.

QRT-PCR analyses of the key genes involved in the anthocyanin biosynthesis pathway were shown in Figure 5, such as drought, salt stress, and heavy metal stress, by maintaining cellular osmotic balance [49,50]. Under Pb stress, the proline concentration in both the LJ43 and ZJ tea plants increased significantly, rising by 108.10% and 39.51%, respectively, compared to the control (CK). This increase in proline is a typical early response mechanism to stress. However, following treatment with exogenous melatonin in combination with Pb stress, proline concentration in LJ43 and ZJ decreased by 36.36% and 46.23%, respectively, indicating that melatonin has a significant regulatory effect on proline levels. Under Pb stress, soluble protein concentration increased by 8.61% in LJ43, while no significant change was observed in ZJ. After melatonin application, soluble protein concentration increased by 4.12% in LJ43 and 4.61% in ZJ, compared to the Pb-only treatment.

Under Pb stress, the expression levels of these genes were generally downregulated, except for *CsANR1*, *CsLAR*, and *CsC4H* in LJ43 and *CsCHI* and *CsC4H* in ZJ. Compared to Pb treatment, exogenous melatonin (MT+Pb) generally upregulated the expression levels of these genes both in LJ43 and ZJ. However, *CsANR1* and *CsANS* were downregulated in LJ43. Phenylalanine begins its transformation via *CsPAL* and two distinct branches emerge at the point of leucocyanidin, one tea variety might enhance its Pb stress tolerance through anthocyanin synthesis by *CsANS,* while the other one might enhance its Pb stress tolerance through the formations of catechin and epicatechin by *CsLAR*, *CsANR1*, and *CsANR2*. In comparison to the Pb treatment, the upregulation fold changes of *CsPAL* were 5.74 and 2.01 in LJ43 and ZJ under the MT+Pb treatment, respectively. Compared to the Pb treatment, *CsANS* was upregulated in ZJ, but it was downregulated in LJ43 under the MT+Pb treatment. Compared to the Pb treatment, *CsLAR*, *CsANR1*, and *CsANR2* were all upregulated in ZJ under the MT+Pb treatment, with upregulation fold changes of 18.8, 1.26 and 2.99, respectively. Compared to the Pb treatment, *CsLAR* and *CsANR2* were upregulated in LJ43 under the MT+Pb treatment, with upregulation fold changes of 1.26 and 9.11, respectively. However, *CsANR2* was downregulated in LJ43 under the MT+Pb treatment, with a downregulation fold change of 2.43. These indicated that, under Pb stress, exogenous melatonin promoted the synthesis of phenylalanine and facilitated its conversion into both anthocyanins and catechins for ZJ. However, exogenous melatonin reduced the phenylalanine accumulation and favored its conversion into catechins for LJ43 (s, such as drought, salt stress, and heavy metal stress, by maintaining cellular osmotic balance [49,50]. Under Pb stress, the proline concentration in both the LJ43 and ZJ tea plants increased significantly, rising by 108.10% and 39.51%, respectively, compared to the control (CK). This increase in proline is a typical early response mechanism to stress. However, following treatment with exogenous melatonin in combination with Pb stress, proline concentration in LJ43 and ZJ decreased by 36.36% and 46.23%, respectively, indicating that melatonin has a significant regulatory effect on proline levels. Under Pb stress, soluble protein concentration increased by 8.61% in LJ43, while no significant change was observed in ZJ. After melatonin application, soluble protein concentration increased by 4.12% in LJ43 and 4.61% in ZJ, compared to the Pb-only treatment.

In addition, the expressions levels of some genes involved in the heavy metal transportations in tea plants were analyzed. Under Pb stress, the expression levels of these genes were generally upregulated. In general, the expression levels of these genes in LJ43 were higher than those in ZJ under Pb stress. The addition of melatonin significantly reduced the expression levels of these genes both in LJ43 and ZJ. In comparison to the Pb treatment, the downregulation fold changes of *CsMTP1*, *CsMRP3*, *CsZIP1*, *CsHMA5*, *CsCDF1*, and *CsPDR12* were 1.63, 3.44, 29.00, 3.11, 1.92, and 4.85 in ZJ under the MT+Pb treatment, respectively. In comparison to the Pb treatment, the downregulation fold changes of *CsMTP1*, *CsMRP3*, *CsZIP1*, *CsHMA5*, *CsCDF1*, and *CsPDR12* were 1.18, 2.05, 1.51, 3.57, 2.89, and 3.47 in LJ43 under the MT+Pb treatment, respectively. The addition of melatonin significantly downregulated the expression of *CsZIP1* in ZJ under Pb stress.

### 2.5. Melatonin Reduced Pb Accumulation in Tea Leaves in the Field

The surface soil characteristics from the field experiment indicate that the soil pH is 5.4, which is weakly acidic and suitable for tea tree cultivation. The total nitrogen content is 0.1%, and the organic matter content is also 0.1%. The available phosphorus content is 4.3 mg kg^−1^, which is relatively low. The available potassium content is 211.8 mg kg^−1^, sufficient to meet the needs of most crops. Notably, the exchangeable magnesium content in the soil is as high as 915.3 mg kg ^−1^. Since tea trees are typical magnesium-loving plants, this is highly beneficial for their growth (Table 1).

The effect of melatonin on tea plants under Pb stress was evaluated through outdoor field trials. Under normal field conditions (CKs), exogenous melatonin (MT) significantly promoted the aboveground growth of tea plants. However, under Pb stress, the growth of both tea varieties was inhibited to varying degrees. Notably, melatonin application under Pb stress (MT+Pb) alleviated this growth inhibition, with the ZJ variety exhibiting a more pronounced recovery. Regarding Pb accumulation, both tea varieties accumulated Pb in their leaves and stems under normal field conditions (CKs), with LJ43 exhibiting higher levels. Although exogenous melatonin (MT) slightly reduced Pb accumulation under these conditions, the effect was not significant. Under Pb stress, LJ43 rapidly accumulated Pb, with Pb concentration in its leaves 6.67 times higher and in its stems 6.23 times higher than in the ZJ variety. However, melatonin application under Pb stress effectively reduced Pb accumulation in both varieties. Specifically, in the leaves, the Pb concentration in LJ43 under MT+Pb treatment was 31.80% lower than under Pb treatment, while in ZJ it was 29.92% lower. In the stems, MT+Pb treatment resulted in a 12.96% reduction in Pb concentration for LJ43 and a 41.66% reduction for ZJ compared to Pb treatment (Figure 6B,C).

## 3. Discussion

In the past few decades, the extent of Pb pollution in the environment caused by human activity has become quite evident. Due to the scarcity of high-quality water, farmers have been forced to use industrially polluted water for irrigation, which has led to a decrease in crop productivity. Prolonged use of industrial wastewater has resulted in the accumulation of heavy metals in the soil, which, through the food chain, has entered the human body and caused symptoms such as neurological disorders, cardiovascular diseases, and developmental issues in infants [54]. Despite many countries implementing measures to control Pb emissions, global Pb production continues to increase [55]. According to previous studies, more than 32% of tea samples collected nationwide have Pb levels exceeding the national maximum allowable concentration of 2.0 mg kg^−1^ [56]. In recent years, melatonin has been extensively studied as an effective strategy for plants to alleviate abiotic stress [48,57,58]. However, the mechanisms by which melatonin alleviates Pb stress in tea plants remain unclear. This study suggests that exogenous melatonin could improve the Pb stress resistance of tea plants mainly by inducing anthocyanin biosynthesis to enhance the antioxidant defenses. Meanwhile, melatonin might be involved in regulating Pb transport genes to inhibit Pb accumulation in tea plants. Moreover, we also found that the above effects of melatonin were more pronounced in the tea plant variety ZJ, which contains high baseline anthocyanin levels. Our research indicates that exogenous melatonin could serve as a useful strategy for reducing Pb levels in tea plants, ensuring the quality of tea products and reducing tea contamination.

Plants often produce excessive reactive oxygen species (ROS) under abiotic stress, leading to lipid peroxidation [59]. Consistent with the response of other plants under Pb stress [60], tea plants produce excessive H_2_O_2_ and O_2_^−^, leading to lipid peroxidation and an increase in MDA concentration. Additionally, proline rapidly accumulates in the plant as a response to oxidative stress, helping to protect cell structure and function [61]. Plants possess enzymatic and non-enzymatic antioxidants to mitigate oxidative damage. Enzymatic antioxidants include SOD, POD, and CAT, while non-enzymatic antioxidants include polyphenols, flavonoids, glutathione, and ascorbic acid (Figure 7) [62,63]. Melatonin can also act as an antioxidant, directly scavenging ROS and enhancing plant stress resistance by improving antioxidant capacity and secondary metabolite concentration under abiotic stress [35,64]. For example, Jahan et al. [65] found that melatonin increased antioxidant enzyme activity, inhibited ROS production, and elevated phenolics, flavonoids, and anthocyanin concentration in tomatoes under nickel stress. For tea plants, Li et al. [66] indicated that melatonin-enhanced tolerance to As stress was largely dependent on the basal levels of anthocyanin in tea plants. Consistent with previous studies, melatonin reduced ROS and MDA levels in tea plants under Pb stress, enhanced antioxidant enzyme activity, promoted the accumulation of secondary metabolites, and stimulated anthocyanin biosynthesis (Figure 2). It is well known that anthocyanins can scavenge ROS to protect plants from oxidative stress and act as metal chelators to reduce heavy metal toxicity, making them excellent non-enzymatic antioxidants [67,68,69].

Anthocyanins and catechins not only act as ROS scavengers but also serve as oxidative signaling regulators and metal chelators, effectively mitigating oxidative stress caused by heavy metal exposure. Previous studies have shown that melatonin can enhance the accumulation of anthocyanins and catechins in various plants, including tea plants, by promoting the expression of the *ANS* and *LAR* genes [42,53,66,70]. In this study, exogenous melatonin significantly promoted phenylalanine accumulation—a key precursor in anthocyanin biosynthesis—in the ZJ cultivar under lead (Pb) stress. Along with qPCR results and increased anthocyanin and catechin levels, this suggested melatonin might enhance phenylalanine conversion into flavonoids, consistent with previous findings on its role in regulating flavonoid biosynthesis [71]. In contrast, melatonin reduced phenylalanine levels in LJ43, while it promoted its conversion mainly towards catechin synthesis.

Under non-stress conditions (MT vs. CK), despite significantly elevated endogenous melatonin, anthocyanin, and catechin levels remained stable. This might reflect sufficient baseline flavonoid levels under normal physiology and the high energy cost of flavonoid biosynthesis, typically induced by stress [72]. Melatonin also conferred stronger Pb tolerance in ZJ, likely due to its higher basal anthocyanin content and greater responsiveness in flavonoid synthesis, enhancing antioxidant capacity [67]. However, current data were insufficient to determine whether melatonin-mediated Pb tolerance depends more on anthocyanin or catechin accumulation. Both are effective in scavenging ROS and chelating metal ions [45], but differ in localization, accumulation patterns, and physiological roles. Further studies are needed to identify which plays the dominant role in melatonin-induced heavy metal tolerance

Previous studies have shown that exogenous melatonin does not improve arsenic (As) tolerance in the ZJ genotype but significantly enhances arsenic tolerance in the LJ43 genotype [66]. This discrepancy may arise from the distinct physicochemical properties and absorption mechanisms of arsenic and Pb within plants. Arsenic primarily exists in the environment as arsenate or arsenite, which are structurally similar to phosphate and are often absorbed by plants through phosphate transporters [73,74]. In contrast, lead is primarily absorbed by plants in its cationic form, typically via calcium ion channels, and tends to bind to cell wall components, thus limiting its mobility within the plant [75]. Furthermore, arsenic and lead differ significantly in their redox activity and toxicity mechanisms, potentially inducing varying levels of oxidative stress and activating different plant defense pathways. The genotype-specific responses observed may be closely related to variations in the expression of plant transport proteins, antioxidant enzyme activities, and the accumulation of flavonoids and other secondary metabolites.

Plants can absorb heavy metals through both leaves and roots, accumulating them within their tissues, with root uptake being the primary pathway for heavy metal entry into plants [76]. Therefore, reducing root absorption of heavy metals is crucial to minimizing heavy metal accumulation in plants. This study found that the application of exogenous melatonin significantly reduced Pb concentration in both tea plant cultivars, which was consistent with the previous findings in other crops [77,78,79]. This indicates that melatonin can inhibit Pb ion uptake and transport in tea plants. QRT-PCR analysis revealed that melatonin treatment significantly downregulated the expression of genes related to heavy metal transport, with the expression of the *CsZIP1* gene in the ZJ cultivar showing the most pronounced decrease after melatonin application (s, such as drought, salt stress, and heavy metal stress, by maintaining cellular osmotic balance [49,50]. Under Pb stress, the proline concentration in both the LJ43 and ZJ tea plants increased significantly, rising by 108.10% and 39.51%, respectively, compared to the control (CK). This increase in proline is a typical early response mechanism to stress. However, following treatment with exogenous melatonin in combination with Pb stress, proline concentration in LJ43 and ZJ decreased by 36.36% and 46.23%, respectively, indicating that melatonin has a significant regulatory effect on proline levels. Under Pb stress, soluble protein concentration increased by 8.61% in LJ43, while no significant change was observed in ZJ. After melatonin application, soluble protein concentration increased by 4.12% in LJ43 and 4.61% in ZJ, compared to the Pb-only treatment.

These transporters play a critical role in heavy metal translocation [80,81,82,83]. Consistent with previous studies, melatonin reduced cadmium concentration in cotton roots and leaves by downregulating cadmium ion transporter gene expression [84]. Furthermore, earlier studies indicate that melatonin can chelate heavy metal ions, forming stable complexes that decrease heavy metal accumulation in plants and sequester heavy metal ions in the plant cell wall or vacuole [85,86]. Melatonin can also reduce Pb upward transport from roots to shoots, thereby limiting Pb migration and lowering Pb concentrations in tea plants [87]. These findings suggest that melatonin’s anti-Pb effects not only directly regulate Pb uptake but also involve plant water–salt balance, ion regulation, and other stress resistance mechanisms.

## 4. Materials and Methods

### 4.1. Plant Species and Experimental Design

#### 4.1.1. Hydroponic Experiment

The experimental materials consisted of two tea cultivars, ZJ and LJ43. Tea plants of the same age, free from pests and diseases, and in healthy growth conditions were selected as experimental subjects. Root treatment: The roots of LJ43 and ZJ were thoroughly washed and grown in water for one week. They were then transferred to a 1/8-strength nutrient solution provided by COOLABER SCIENCE & TECHNOLOGY (Beijing, China) for two weeks, with the pH adjusted to 5 using 0.1 mol L^−1^ HCl and NaOH until new roots developed. The full-strength nutrient solution contained macronutrients (mg L^−1^) NH_4_NO_3_ (80.04), KH_2_PO_4_ (8.53), K_2_SO_4_ (52.27), MgSO_4_ (80.64), CaCl_2_ (58.82), Al (SO_4_)_3_·8H_2_O (23.33), and EDTA-FeNa (1.77), and micronutrients (mg L^−1^) H_3_BO_3_ (0.43), ZnSO_4_·7H_2_O (0.19), MnSO_4_·H_2_O (0.17), (NH_4_) Mo_7_O_24_·4H_2_O (0.058), and CuSO_4_·5H_2_O (0.03). When the new roots developed, tea plants with uniform growth and development were selected and transferred to a 1/4-strength nutrient solution. The plants were secured using sponges placed in holes on the bucket lids. Each treatment included three biological replicates with seven plants per replicate. Then, the following four treatments were performed:CK: Foliar spraying with pure water, no Pb treatment applied to the roots;MT: Foliar spraying with a 100 μmol L^−1^ melatonin solution, no Pb treatment applied to the roots;Pb: Foliar spraying with pure water, and root treatment with a nutrient solution containing a Pb concentration of 0.29 mmol L^−1^;MT+Pb: Foliar spraying with a 100 μmol L^−1^ melatonin solution, and root treatment with a nutrient solution containing a Pb concentration of 0.29 mmol L^−1^.

Based on our previous results, we determined that 0.29 mmol L^−1^ was the optimal Pb stress concentration. This concentration was high enough to induce stress in the tea plants without causing their death, while also being sufficiently effective to produce a meaningful stress response. In the experimental group, tea seedlings were sprayed with 100 µmol L^−1^ melatonin on their leaves every evening. Pb acetate treatment began 24 h after the initial melatonin application. The nutrient solution was prepared with 0.29 mmol L^−1^ Pb acetate and adjusted to a pH of 5. The Pb stress treatment lasted for 10 days. During this period, one-quarter of the nutrient solution was replaced every three days with a fresh solution containing Pb. At the end of the treatment, roots, stems, and mature leaves of the tea plants were collected for subsequent analyses.

#### 4.1.2. Field Experiment

The experiment was conducted on a tea plantation in Shaoxing, Zhejiang Province, China (30.03° N, 120.58° E), where all tea plants were managed under the same cultivation practices. Two tea cultivars, ZJ and LJ43, with similar growth vigor, were selected for the study. After sunset, the tea plants were treated with a foliar spray of 100 μmol L^−1^ melatonin solution. After 24 h, the experimental group was subjected to Pb stress using a lead acetate solution. Considering previous studies indicating that Pb in soil can form complexes that reduce its toxicity, a higher concentration of 1.93 mmol L^−1^ Pb solution was applied via irrigation to ensure the soil was fully saturated to a depth of 20 cm [88]. The Pb solution was administered once per week for two consecutive weeks, with treatment timing determined based on the one-bud, two-leaf growth stage. After treatment, samples were collected from the top bud to the third or fourth fully expanded leaf and immediately transported on dry ice to the laboratory for further analysis.

The four treatments were as follows:CK: Foliar spraying with pure water, no Pb treatment applied to the roots.MT: Foliar spraying with 100 μmol L^−1^ melatonin solution, no Pb treatment applied to the roots.Pb: Foliar spraying with pure water, Pb solution (1.93 mmol L^−1^) applied to the roots.MT+Pb: Foliar spraying with 100 μmol L^−1^ melatonin solution, Pb solution (1.93 mmol L^−1^) applied to the roots.

### 4.2. Measurement Methods

Pb concentration: The tea tree leaves, stems, and roots were freeze-dried, ground, and 0.05 g of the sample was weighed into digestion tubes. Subsequently, 4 mL of HNO_3_ solution was added, and the mixture was left to stand overnight. The samples were then placed in a metal digestion apparatus (anon SH230N) (Hanon Advanced Technology Group Co., Ltd., Jinan, China) and digested at a controlled temperature of 130 °C until white fumes appeared and the digestion solution became colorless and transparent [89]. The digested solution was diluted to 10 mL with deionized water for analysis. The digestate was determined using inductively coupled plasma mass spectrometry (ICP-MS) (PlasmaQuant MS Elite) (Analytik Jena GmbH + Co. KG, Jena, Germany) [70].

Endogenous melatonin concentration: A 0.2 g sample of tea tree leaf was weighed and ground into a fine powder using liquid nitrogen. The powdered sample was then extracted under precooled conditions with 1.0 mL of chromatographic-grade methanol. After ultrasonic treatment and centrifugation at 12,000 rpm for 10 min at 4 °C, the supernatant was collected in a fresh tube and dried under a nitrogen stream. The residue was then dissolved in a 1.0 mL methanol–water mixture (1:1, *v*/*v*), followed by centrifugation at 12,000 rpm for 10 min at 4 °C. The resulting solution was filtered through a 0.22 µm organic nylon membrane and analyzed using a SCIEX ultra-performance liquid chromatography-tandem triple quadrupole mass spectrometer system (Triple Quad^TM^ LC-MS/MS 5500+) (SCIEX, Fremont, CA, USA) [90].

Polyphenols and flavonoid concentration: A 0.05 g freeze-dried tea sample was placed into a 2 mL centrifuge tube, and 1 mL of pre-chilled 80% (*v*/*v*) methanol was added. The mixture was sonicated for 15 min, followed by centrifugation at 12,000 rpm for 10 min at 4 °C. The supernatant was then used for the determination of total phenols and flavonoid concentration. For the determination of phenols concentration, 10 µL of the supernatant was taken and mixed sequentially with 90 µL of 80% (*v*/*v*) methanol, 2 mL of deionized water, and 0.1 mL of Folin–Ciocalteu reagent. After a thorough mixing, 0.75 mL of 26.7% Na_2_CO_3_ solution and 1 mL of deionized water were added. The mixture was incubated in the dark at room temperature for 1 h. Absorbance was then measured at 760 nm using a Tecan Austria GmbH A-5082 spectrophotometer (Tecan Austria GmbH, Grödig, Austria) [91].

For the determination of flavonoid concentration, 0.01 mL of the supernatant was mixed with 0.19 mL of 80% (*v*/*v*) methanol in a 5 mL centrifuge tube. Then, 1.3 mL of 30% (*v*/*v*) ethanol was added, followed by 75 µL of 5% NaNO_2_ solution. After incubating for 5 min, 75 µL of 10% Al (NO_3_)_3_ solution was added and left to react for 6 min. Subsequently, 0.5 mL of 5% NaOH solution was added, followed by 0.85 mL of 30% ethanol. The mixture was thoroughly mixed and allowed to stand at room temperature for 10 min. The absorbance was then measured at 510 nm using a Tecan Austria GmbH A-5082 microplate reader (Tecan Austria GmbH, Grödig, Austria) [92].

Anthocyanins concentration: A total of 0.05 g of freeze-dried tea leaf sample was weighed and mixed with 1 mL of an anthocyanin extraction solution (prepared by combining 1 mL of hydrochloric acid, 18 mL of propanol, and 81 mL of deionized water to make a total of 100 mL). The mixture was incubated in a water bath and extracted overnight in the dark while being shaken at 120 rpm. After extraction, the mixture was centrifuged at room temperature and the supernatant collected. The absorbance was then measured at 535 nm and 650 nm using a microplate reader (Tecan Austria GmbH A-5082) (Tecan Austria GmbH, Grödig, Austria) [93].

Relative chlorophyll concentration: After 1, 4, 7, and 10 days of the treatment, the relative chlorophyll concentration of the same mature leaf was measured using a portable SPAD meter (SPAD-502PLUS) (Konica Minolta, Tokyo, Japan.), with each measurement repeated four times [94]. Each treatment included 3 biological replicates with 7 plants per replicate.

The soluble protein concentration was measured using the Coomassie Brilliant Blue G-250 method. Proline was extracted using a 3% sulfosalicylic acid solution, and its concentration was determined by the acid ninhydrin method. Malondialdehyde (MDA) was measured using a 0.5% thiobarbituric acid solution [95]. The hydrogen peroxide concentration was determined using titanium sulfate, with a standard curve made using 30% analytical grade H_2_O_2_ [96]. The uperoxide anion concentration was measured using p-aminobenzenesulfonic acid and α-naphthylamine, with NaNO_2_ used to prepare the standard curve. Superoxide dismutase (SOD) was measured using the method of Ukeda, et al. [97], catalase (CAT) using the method of Johansson [98], and peroxidase (POD) using the method of Reuveni [99].

Gene expression analysis (qRT-PCR): RNA was extracted from tea leaves using an RNA extraction kit (Sangon Biotech) (Shanghai, China). First-strand cDNA was synthesized using a reverse transcription kit (Vazyme) (Nanjing, China.). RT-qPCR was performed on a real-time PCR system (CFX96^TM^ Real-Time System) (Hercules, CA, USA) using a ChamQ SYBR Color qPCR Master Mix (Vazyme) (Nanjing, China). The relative expression levels of key genes involved in Pb transport and anthocyanin biosynthesis were quantified, with GAPDH used as the reference gene. The fold change in gene expression was calculated using the 2^−ΔΔCt^ method. The primer design for quantitative fluorescence PCR is shown in Appendix A.

### 4.3. Statistical Analysis

All data were subject to analysis of variance (ANOVA) using DPS 7.0 software package. Three independent biological replicates were analyzed, and the means were compared using a LSD’s test at the *p* < 0.05 level.

## 5. Conclusions

In conclusion, this study demonstrates that exogenous melatonin application effectively reduces Pb toxicity in tea plants by decreasing Pb accumulation and mitigating oxidative stress. These effects may be partially attributed to enhanced flavonoid metabolism, including the biosynthesis of anthocyanins and catechins which contribute to the antioxidant defense system and support Pb stress tolerance. However, the extent of melatonin-induced tolerance varies between tea genotypes, likely due to differences in basal anthocyanin levels and metabolic responses. Moreover, melatonin was involved in the regulation of Pb transporter genes, with a significant downregulation of *CsZIP1* potentially playing a critical role in restricting Pb uptake in both roots and shoots. Although previous studies support the involvement of flavonoids in melatonin-mediated stress resistance, the specific regulatory relationship between melatonin-induced flavonoid biosynthesis and Pb transporter gene expression remains to be elucidated. Overall, this study provides insights into the physiological and molecular mechanisms underlying melatonin-enhanced Pb stress tolerance in tea plants, offering a promising, environmentally friendly strategy to improve tea crop safety and quality in contaminated environments.

## Figures and Tables

**Figure 1 plants-14-01417-f001:**
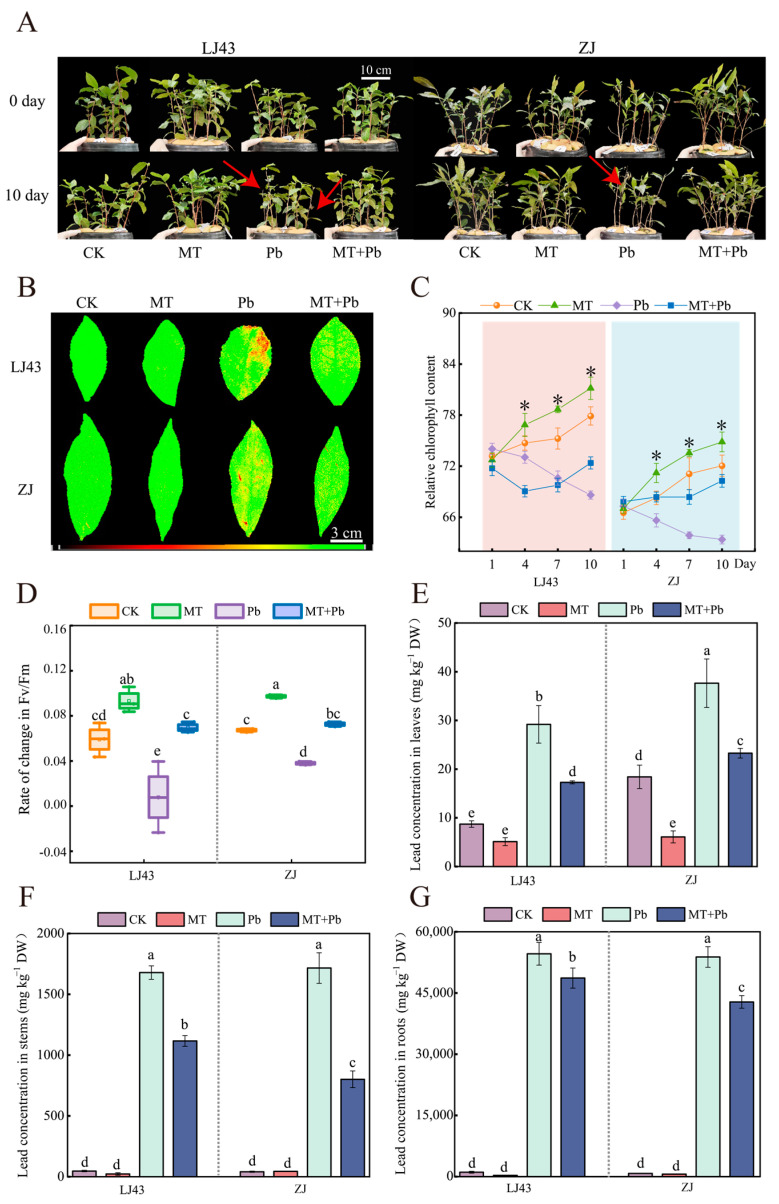
Exogenous melatonin improved the photosynthetic capacity of LJ43 and ZJ tea plants under Pb stress and reduced the Pb concentration in tea plants. (**A**) Phenotypic images before and after 10 days of Pb treatment, with arrows highlighting the withered and yellowing leaves; (**B**) chlorophyll fluorescence images after 10 days of Pb treatment. The color gradient indicates the degree of leaf stress, with green representing healthy tissue, yellow indicating moderate stress, and red showing severe damage or necrosis; (**C**) changes in relative chlorophyll concentration of tea plants within 10 days. Asterisks indicate significant differences between CK and MT, and between Pb and MT+Pb; (**D**) changes in the Fv/Fm ratio; (**E**) lead concentration in leaves; (**F**) Pb concentration in stems; (**G**) Pb concentration in roots. The vertical line in the figure is used to distinguish the data of different cultivars. Data are the mean of three replicates (±standard deviation, SD), and means followed by the same letters are not significantly different according to LSD’s test (*p* < 0.05).

**Figure 2 plants-14-01417-f002:**
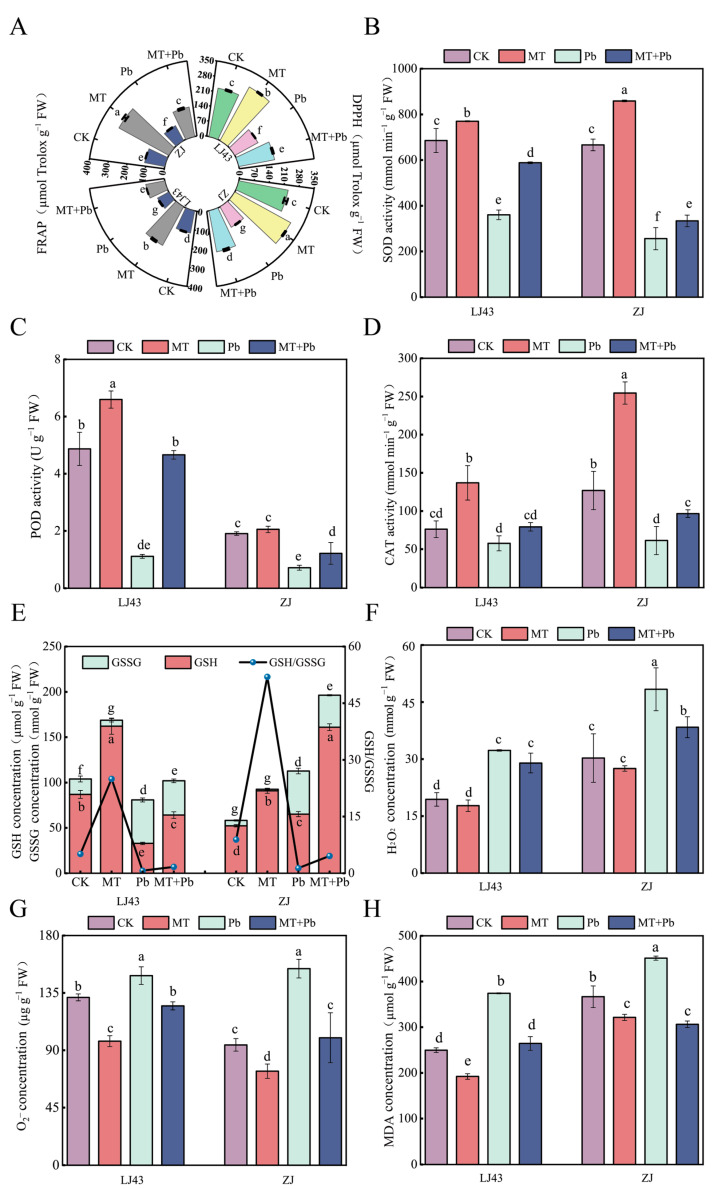
Exogenous melatonin enhanced the antioxidant capacity of the leaves of LJ43 and ZJ tea plants under Pb stress and reduced the levels of reactive oxygen species (ROS). (**A**) Total antioxidant capacity: DPPH and FRAP; (**B**) SOD activity; (**C**) POD activity; (**D**) CAT activity; (**E**) glutathione redox status and ratio; (**F**) hydrogen peroxide concentration; (**G**) superoxide anion concentration; (**H**) malondialdehyde (MDA) concentration. Data are the mean of three replicates (±standard deviation, SD), and means followed by the same letters are not significantly different according to LSD’s test (*p* < 0.05).

**Figure 3 plants-14-01417-f003:**
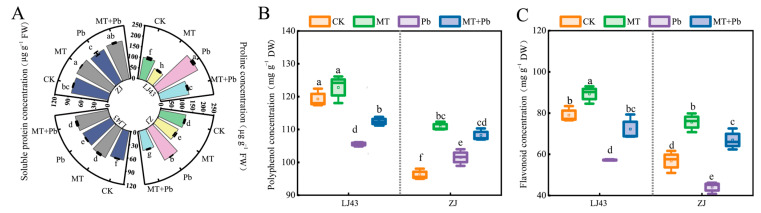
Exogenous melatonin can enhance the levels of osmotic adjustment substances and secondary metabolites in LJ43 and ZJ tea plant leaves under Pb stress. (**A**) Proline and soluble protein concentration; (**B**) polyphenol concentration; (**C**) flavonoid concentration. The vertical line in the figure is used to distinguish the data of different cultivars. Data are the mean of three replicates (±standard deviation, SD), and means followed by the same letters are not significantly different according to LSD’s test (*p* < 0.05).

**Figure 4 plants-14-01417-f004:**
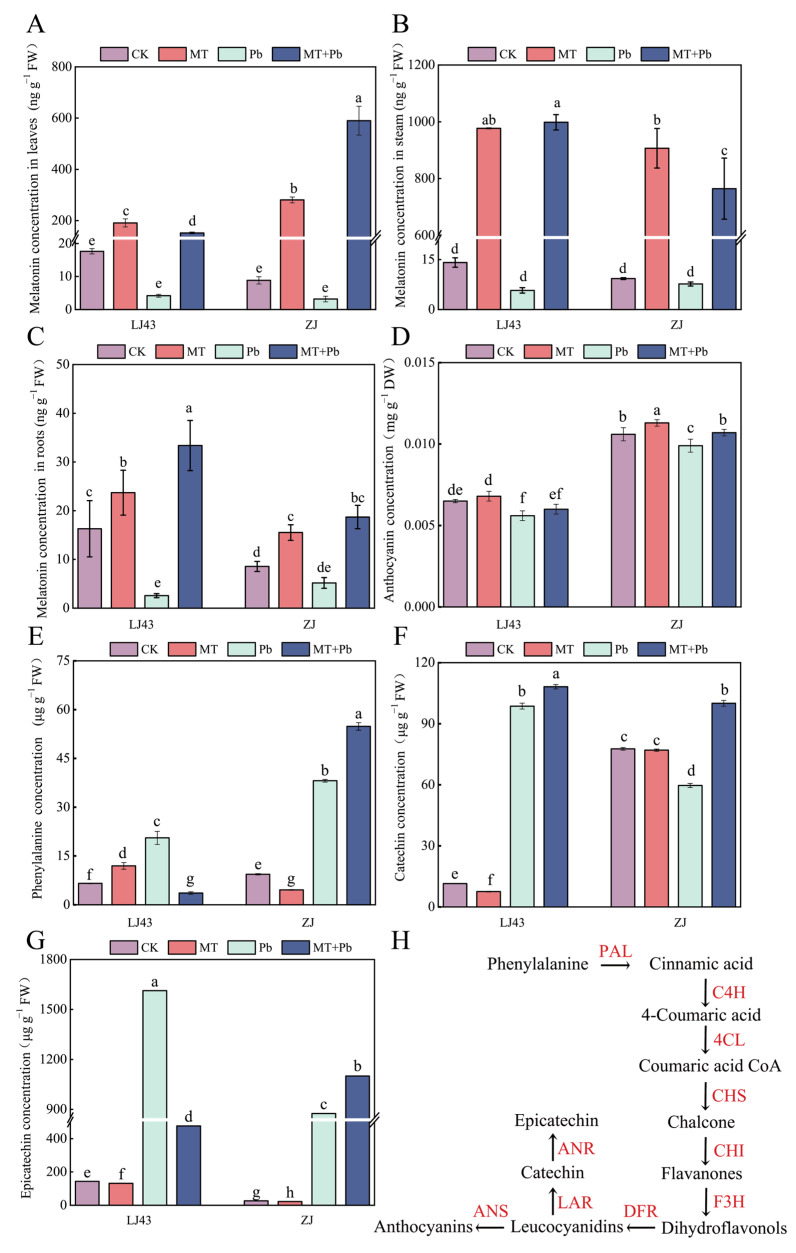
Exogenous melatonin can enhance the concentrations of endogenous melatonin in the roots, stems, and leaves of LJ43 and ZJ, as well as key metabolites involved in the anthocyanin biosynthesis pathway. (**A**) Melatonin concentration in leaves; (**B**) melatonin concentration in stems; (**C**) melatonin concentration in roots; (**D**) anthocyanin concentration; (**E**) phenylalanine concentration; (**F**) catechin concentration; (**G**) epicatechin concentration; (**H**) anthocyanin biosynthesis pathway [53]. The white line in the figure represents gaps in the axes. Data are the mean of three replicates (±standard deviation, SD), and means followed by the same letters are not significantly different according to LSD’s test (*p* < 0.05).

**Figure 5 plants-14-01417-f005:**
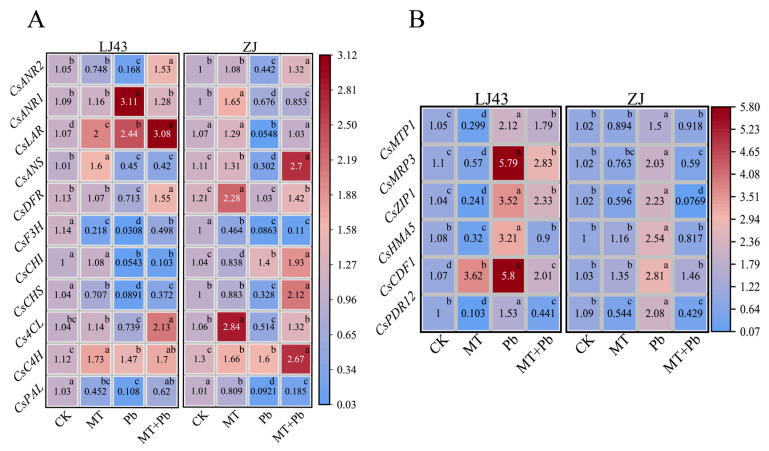
Expression analysis of anthocyanin biosynthesis-related genes and heavy metal transporter genes in tea plants, the scale bar in the figure represents 2^−ΔΔCt^. (**A**) Expression levels of genes related to the anthocyanin biosynthesis pathway; (**B**) expression levels of Pb ion transport genes. Data are the mean of three replicates, and no significant difference is indicated by the same letters according to the LSD test (*p* < 0.05).

**Figure 6 plants-14-01417-f006:**
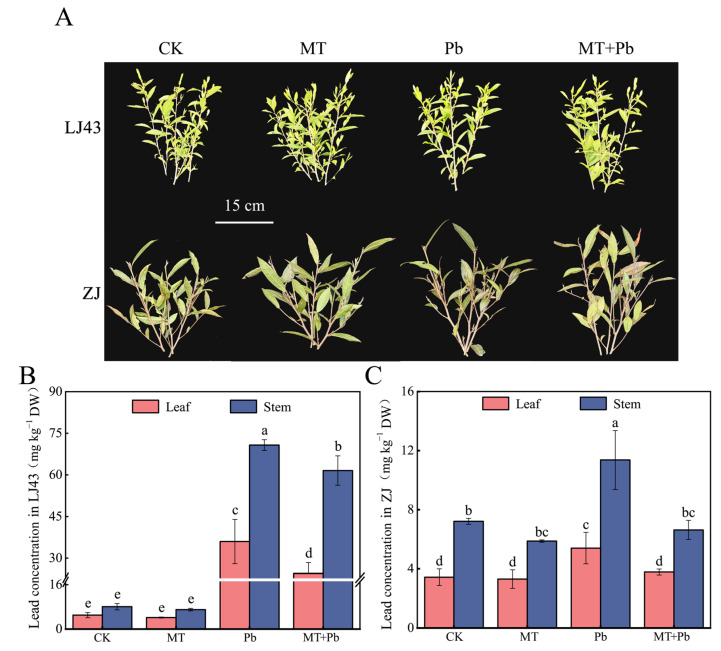
The alleviating effect of exogenous melatonin on LJ43 and ZJ under 1.93 mmol L^−1^ Pb stress in field trials. (**A**) Phenotypic changes in tea plants after different Pb stress treatments; (**B**) changes in Pb concentration in the leaves and stems of LJ43 with exogenous melatonin treatment; (**C**) changes in Pb concentration in the leaves and stems of ZJ with exogenous melatonin treatment. The white line in the figure represents gaps in the axes. Data are the mean of three replicates (±standard deviation, SD), and means followed by the same letters are not significantly different according to LSD’s test (*p* < 0.05).

**Figure 7 plants-14-01417-f007:**
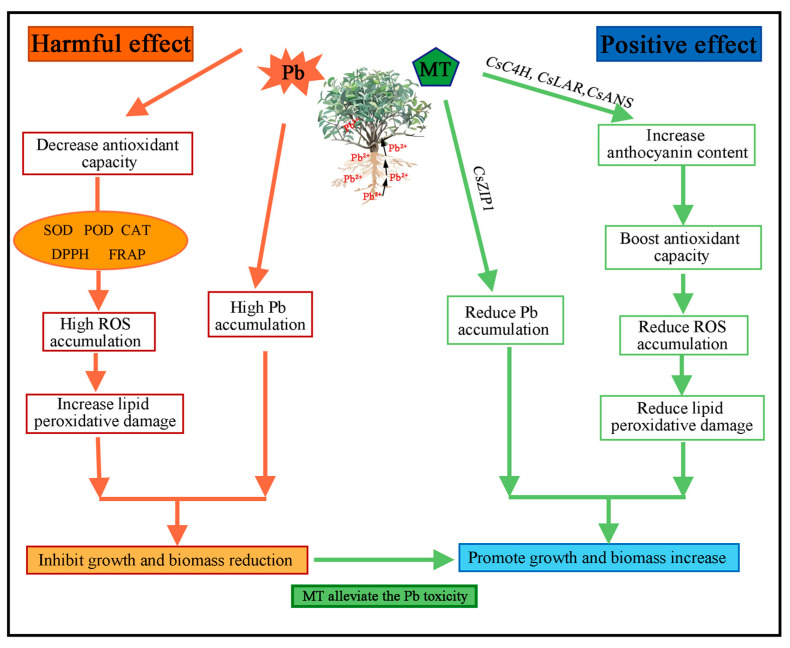
Mechanism of exogenous melatonin alleviating Pb stress in tea plants: promoting anthocyanin synthesis, enhancing antioxidant capacity, and reducing Pb accumulation. The orange arrows indicate the harmful effects induced by Pb, while the green arrows represent the positive effects of melatonin (MT) in alleviating Pb toxicity.

**Table 1 plants-14-01417-t001:** Field experiment surface soil properties table.

pH	Total Nitrogen (%)	Organic Matter (%)	Available Phosphorus (mg kg^−1^)	Available Potassium (mg kg^−1^)	Exchangeable Magnesium (mg kg^−1^)
5.4	0.1	0.1	4.3	211.8	915.3

## Data Availability

All data are included in this paper. Additional information can be provided upon request to the correspondence author.

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
