# Peer review of "Melatonin-Mediated Regulation of Antioxidant Defense Enhances the Resistance of Tea Plants (*Camellia sinensis* L.) to Lead-Induced Stress"

_plants, 2025, doi:10.3390/plants14101417_

Round 1

Reviewer 1 Report

Comments and Suggestions for Authors

The accumulation of lead in plants can harm human health through the food chain. How to reduce lead accumulation in plants through scientific means has always been a research hotspot. Melatonin, a hormone widely distributed in plants, plays an important role in plant resistances to various stresses. This is an attractive research that traced new method of reducing lead accumulations in tea leaves. The results are conducive to understand the influence of melatonin on alleviating the lead threat, and provide scientific reference for ensuring the green, healthy and sustainable development of tea plants. The manuscript was well organized and can be accepted for publication after moderate revision. Some specific issues are listed as follows:

  1. Line 87-88: “Melatonin has been widely studied for its role in enhancing plant tolerance to various abiotic stresses..” This sentence needs to be supported by Reference. Such as Chakraborty, S., Raychaudhuri, S.S. Melatonin improves the lead tolerance in Plantago ovata by modulating ROS homeostasis, phytohormone status and expression of stress-responsive genes. Plant Cell Rep 44, 39 (2025). https://doi.org/10.1007/s00299-025-03424-x
  2. Line 148-149: “POD activity increased by 300% and 71.04%, and CAT activity increased by 37.30% and 57.50%, respectively.” Keep all data formats consistent.
  3. I suggest to increase the basic properties of tea garden soil, such as pH and organic matter content, in “4.2.2 Field Experiment”.
  4. The scale bar should be added to Figure 1B.
  5. Lines 134, 178, 205, 224, 284, 308: “(P < 0.05)” needs to be in italics.
  6. The title of Figure 1E is confusing and needs to be revised.
  7. There are multiple instances throughout the text where “content” and “concentration” are confused and need to be reorganized (Figure 3, Figure 4 and so on).
  8. Because previous studies have investigated As accumulation in tea plants, but the results seem to differ from the present study, it is necessary to explain the differences in plant accumulation of As and Pb in the text.
  9. Anthocyanin biosynthesis pathway in Figure 4F needs to be supported by references.
  10. What does the scale bar in Figure 5 represent? It needs to be labeled.
  11. All gene names in Figure 7 and throughout the text need to be italicized.

Author Response

Thank you very much for giving us the opportunity to revise our manuscript (plants-3583606).

We have studied reviewer’s comments and suggestions carefully and have made revision accordingly which are marked in red in the manuscript. We have also replied point by point to reviewer’s comments in the following letter which are marked in blue. Attached please find the revised version, which we would like to submit for your kind consideration.

We would like to express our great appreciation to you and reviewers for comments on our manuscript. Looking forward to hearing from you.

With best regards!

Yours sincerely,

Li Ruan

Reviewer 1

Comments and Suggestions for Authors

The accumulation of lead in plants can harm human health through the food chain. How to reduce lead accumulation in plants through scientific means has always been a research hotspot. Melatonin, a hormone widely distributed in plants, plays an important role in plant resistances to various stresses. This is an attractive research that traced new method of reducing lead accumulations in tea leaves. The results are conducive to understand the influence of melatonin on alleviating the lead threat, and provide scientific reference for ensuring the green, healthy and sustainable development of tea plants. The manuscript was well organized and can be accepted for publication after moderate revision. Some specific issues are listed as follows:

Comments 1: Line 87-88: “Melatonin has been widely studied for its role in enhancing plant tolerance to various abiotic stresses.” This sentence needs to be supported by Reference. Such as Chakraborty, S., Raychaudhuri, S.S. Melatonin improves the lead tolerance in Plantago ovata by modulating ROS homeostasis, phytohormone status and expression of stress-responsive genes. Plant Cell Rep 44, 39 (2025). https://doi.org/10.1007/s00299-025-03424-x

Response 1: Thank you very much for your valuable suggestion! According to your recommendation, we have inserted two references (lines 103, 688-691):

  1. Chakraborty, S.; Raychaudhuri, S.S. Melatonin improves the lead tolerance in Plantago ovata by modulating ROS homeostasis, phytohormone status and expression of stress-responsive genes. Plant Cell Rep. 2025, 44, 39.
  2. Zhang, N.; Sun, Q.; Zhang, H.; Cao, Y.; Weeda, S.; Ren, S.; Guo, Y.-D. Roles of melatonin in abiotic stress resistance in plants. Exp. Bot. 2015, 66, 647–656.

Comments 2: Line 148-149: “POD activity increased by 300% and 71.04%, and CAT activity increased by 37.30% and 57.50%, respectively.” Keep all data formats consistent.

Response 2: Thank you very much for your suggestion. Following your recommendation, we have standardized the format of all data in lines 165-166, changing "300%" to "300.00%" to ensure consistency.

Comments 3: I suggest to increase the basic properties of tea garden soil, such as pH and organic matter content, in “4.2.2 Field Experiment”.

Response 3: Following your valuable suggestion, we have introduced the basic characteristics of the tea garden soil as follows (lines 301-308):

The surface soil characteristics from the field experiment indicate that the soil pH is 5.4, which is weakly acidic and suitable for tea tree cultivation. The total nitrogen content is 0.1%, and the organic matter content is also 0.1%. The available phosphorus content is 4.3 mg kg-1, which is relatively low. The available potassium content is 211.8 mg kg-1, sufficient to meet the needs of most crops. Notably, the exchangeable magnesium content in the soil is as high as 915.3 mg kg-1. Since tea trees are typical magnesium-loving plants, this is highly beneficial for their growth (Table 1).

Table 1. Field experiment surface soil properties table.

pH

Total

nitrogen (%)

Organic

matter (%)

Available phosphorus (mg kg-1)

Available potassium

(mg kg-1)

Exchangeable magnesium

(mg kg-1)

5.4

0.1

0.1

4.3

211.8

915.3

Comments 4: The scale bar should be added to Figure 1B.

Response 4: Thank you very much for your valuable suggestion! Following your recommendation, we have added the scale bar in Figure 1B in line 141 of the manuscript.

Comments 5: Lines 134, 178, 205, 224, 284, 308: “(P < 0.05)” needs to be in italics.

Response 5: Thank you very much for your valuable suggestion! Following your recommendation, we have italicized "(P < 0.05)" in lines 148, 191, 219, 242, and 299of the manuscript.

Comments 6: The title of Figure 1E is confusing and needs to be revised.

Response 6: Following your suggestion, we have revised the term “Leaf lead content” to “Lead concentration in leaves” in Figure 1E (line 146) to improve clarity and consistency.

Comments 7: There are multiple instances throughout the text where “content” and “concentration” are confused and need to be reorganized (Figure 3, Figure 4 and so on).

Response 7: Thank you very much for your suggestion! Following your recommendation, we have standardized the content in Figures 1, 2, 3, and 4 to concentration, and the corresponding figure legends have also been revised accordingly. The line numbers are 141, 184, 214, and 235.

Comments 8: Because previous studies have investigated As accumulation in tea plants, but the results seem to differ from the present study, it is necessary to explain the differences in plant accumulation of As and Pb in the text.

Response 8: Following your suggestion, we have explained the difference between the effects of exogenous melatonin on arsenic accumulation and lead accumulation in tea plants in lines 397-409 of the manuscript as follows:

Previous studies have shown that exogenous melatonin does not improve arsenic (As) tolerance in the ZJ genotype but significantly enhances arsenic tolerance in the LJ43 genotype (Li et al. 2021). This discrepancy may arise from the distinct physicochemical properties and absorption mechanisms of arsenic and Pb within plants. Arsenic primarily exists in the environment as arsenate or arsenite, which are structurally similar to phosphate and are often absorbed by plants through phosphate transporters (Li et al. 2016; Zhao et al. 2009). In contrast, lead is primarily absorbed by plants in its cationic form, typically via calcium ion channels, and tends to bind to cell wall components, thus limiting its mobility within the plant (Sharma and Dubey 2005). Furthermore, arsenic and lead differ significantly in their redox activity and toxicity mechanisms, potentially inducing varying levels of oxidative stress and activating different plant defense pathways. The genotype-specific responses observed may be closely related to variations in the expression of plant transport proteins, antioxidant enzyme activities, and the accumulation of flavonoids and other secondary metabolites.

Comments 9: Anthocyanin biosynthesis pathway in Figure 4F needs to be supported by references.

Response 9: Thank you very much for your valuable suggestion! Following your recommendation, we have inserted relevant references on the anthocyanin synthesis pathway as follows: 

  1. Ma, Y.; Ma, X.; Gao, X.; Wu, W.; Zhou, B. Light Induced Regulation Pathway of Anthocyanin Biosynthesis in Plants. Int. J. Mol. Sci. 2021, 22, 11116.

Comments 10: What does the scale bar in Figure 5 represent? It needs to be labeled.

Response 10: Thank you very much for your valuable suggestion. We have added an explanation of the scale bar in the figure legend as follows (lines 295-296):

Expression analysis of anthocyanin biosynthesis-related genes and heavy metal transporter genes in tea plants, the scale bar in the figure represents 2-ΔΔCt.

Comments 11: All gene names in Figure 7 and throughout the text need to be italicized.

Response 11: Thank you very much for your valuable suggestion. In accordance with your comments, we have revised the formatting of all gene names by italicizing them in the relevant sections of the manuscript, including lines 20, 23, and 264–293, as well as in Figure 7 (line 429), to ensure consistency and adherence to scientific writing conventions.

Reviewer 2 Report

Comments and Suggestions for Authors

This manuscript investigates the role of melatonin in enhancing lead (Pb) tolerance in two tea genotypes, Longjing43 (LJ43) and Zijuan (ZJ), by improving their antioxidant defense and altering flavonoid synthesis. This study is innovative in investigating the effect of melatonin on tea plants under metal stress. However, the results do not fully support the conclusions, and the correlation between melatonin levels and anthocyanin and catechin synthesis in the tea plant is still unclear. The differences between the two genotypes also lack sufficient explanation. A major revision is recommended, and additional data may be needed to support the conclusion before publication.

Major Issues

  1. Since Pb accumulation and stress are heaviest in the roots, and melatonin was applied via foliar spray, it is suggested that melatonin be applied directly to the root, which might be more effective. In addition, melatonin content was only measured in the leaves; its levels in stems and roots were not assessed. As the root system is most affected and plays an important role in Pb stress response, endogenous melatonin levels in the roots should also be quantified.
  2. In the Discussion part, the authors suggest melatonin may upregulate anthocyanin and catechin biosynthesis to enhance Pb stress tolerance. However, this is not sufficiently supported by the current data. In Figure 1E, melatonin significantly reduces Pb content in ZJ leaves under both normal and Pb-stressed conditions. Also, melatonin content increases significantly with treatment (MT vs. CK, MT+Pb vs. Pb). However, in Figure 4B and 4D, there is only a minimal increase in anthocyanin levels with melatonin treatment in ZJ, and catechin content shows no significant difference in the MT group compared to CK. In LJ43, melatonin decreases leaf Pb content only under Pb stress, but its endogenous melatonin level increases with or without Pb stress. Furthermore, there are no significant differences observed in anthocyanin content between MT+Pb and Pb groups. Overall, endogenous melatonin and Pb levels do not show a strong association with anthocyanin or catechin content, at least in the leaf tissue. It also remains unclear which of these two flavonoids plays a major role in melatonin-induced Pb tolerance.
  3. To better demonstrate the role of melatonin in regulating anthocyanin biosynthesis under Pb stress, a melatonin synthesis inhibitor such as p-chlorophenylalanine is suggested to clarify the endogenous melatonin level in tea plants during Pb stress.

Minor Issues

  1. As the latter half of this study focuses on anthocyanins and catechins, relevant background information should be briefly provided in the Introduction section.
  2. In the last paragraph of the Introduction, please briefly summarize the main findings rather than only stating the research aim.
  3. Lines 139–143, please add figure numbers when describing the results.
  4. Figures 2D–E are missing genotype labels (LJ43 and ZJ).
  5. Lines 193–195, it is stated that polyphenols in ZJ under Pb stress decreased by 5.22% compared to CK. However, Figure 3B shows an increase. Please revise and provide clarification.
  6. Figure 5 legend, there are no statistical comparisons in this figure, as stated in the figure legend. Please add relevant significance markers.
  7. In section 4.2 (Measurement Methods), part 1 on Pb content: Only leaf extraction is described, but Figure 2 also includes data on stem and root Pb content. Please add the extraction details for these tissues.
Comments on the Quality of English Language

There are several typos in the abstract, main text, and figure; please review and revise them.

Author Response

Thank you very much for giving us the opportunity to revise our manuscript (plants-3583606).

We have studied reviewer’s comments and suggestions carefully and have made revision accordingly which are marked in red in the manuscript. We have also replied point by point to reviewer’s comments in the following letter which are marked in blue. Attached please find the revised version, which we would like to submit for your kind consideration.

We would like to express our great appreciation to you and reviewers for comments on our manuscript. Looking forward to hearing from you.

With best regards!

Yours sincerely,

Li Ruan

Reviewer 2

Comments and Suggestions for Authors

This manuscript investigates the role of melatonin in enhancing lead (Pb) tolerance in two tea genotypes, Longjing43 (LJ43) and Zijuan (ZJ), by improving their antioxidant defense and altering flavonoid synthesis. This study is innovative in investigating the effect of melatonin on tea plants under metal stress. However, the results do not fully support the conclusions, and the correlation between melatonin levels and anthocyanin and catechin synthesis in the tea plant is still unclear. The differences between the two genotypes also lack sufficient explanation. A major revision is recommended, and additional data may be needed to support the conclusion before publication.

Major Issues

Comments 1: Since Pb accumulation and stress are heaviest in the roots, and melatonin was applied via foliar spray, it is suggested that melatonin be applied directly to the root, which might be more effective. In addition, melatonin content was only measured in the leaves; its levels in stems and roots were not assessed. As the root system is most affected and plays an important role in Pb stress response, endogenous melatonin levels in the roots should also be quantified.

Response 1: Thank you very much for your valuable comments.

1) Firstly, Pb primarily exists in non-mobile forms in the soil, with the rhizosphere being the most heavily contaminated zone. Therefore, applying melatonin directly to the roots of tea plants may be a more effective strategy for mitigating Pb stress. However, previous studies have shown that the effects of root application tend to be slower and are more susceptible to variation due to soil properties (Madigan et al. 2019; Muhammad et al. 2024). In actual production, if melatonin is directly added to the rhizosphere soil, it may eventually lead to an excessive accumulation of melatonin over a long period of time. The excessive use of melatonin has negative effects on plant growth (Gamalero and Glick 2025). Thus the long-term melatonin accumulations in soils may have unknown impacts on the soil ecosystem. However, foliar application can avoid potential degradation of melatonin by soil microorganisms and minimize soil-related interference (Arnao and Hernández-Ruiz 2015). In addition, numerous studies have successfully adopted foliar spraying of melatonin to alleviate stress in plants (Buttar et al. 2020; Ding et al. 2017; Han et al. 2017).

2) Secondly, following your suggestion, we have added the melatonin concentrations of stems and roots in Figure 4B and 4C and provided the corresponding analysis in the manuscript (lines 222-230) as follows:

The application of exogenous melatonin significantly increased the levels of endogenous melatonin in tea plants, indicating that the exogenous melatonin was con-verted into its endogenous form (Figure 4A-C). In LJ43, the endogenous melatonin concentrations in roots, stems, and leaves under the MT+Pb treatment were 13.01, 173.53, and 36.32 times higher than those under the Pb treatment, respectively. In ZJ, the corresponding increases were 3.61, 99.65, and 185.34 times, respectively. Due to the addition of exogenous melatonin, the increase in endogenous melatonin in the leaves of ZJ was significantly greater than that in LJ43, whereas in roots and stems, the in-crease was more pronounced in LJ43 than in ZJ.

Comments 2: In the Discussion part, the authors suggest melatonin may upregulate anthocyanin and catechin biosynthesis to enhance Pb stress tolerance. However, this is not sufficiently supported by the current data. In Figure 1E, melatonin significantly reduces Pb content in ZJ leaves under both normal and Pb-stressed conditions. Also, melatonin content increases significantly with treatment (MT vs. CK, MT+Pb vs. Pb). However, in Figure 4B and 4D, there is only a minimal increase in anthocyanin levels with melatonin treatment in ZJ, and catechin content shows no significant difference in the MT group compared to CK. In LJ43, melatonin decreases leaf Pb content only under Pb stress, but its endogenous melatonin level increases with or without Pb stress. Furthermore, there are no significant differences observed in anthocyanin content between MT+Pb and Pb groups. Overall, endogenous melatonin and Pb levels do not show a strong association with anthocyanin or catechin content, at least in the leaf tissue. It also remains unclear which of these two flavonoids plays a major role in melatonin-induced Pb tolerance.

Response 2: Thank you very much for your valuable suggestions! Based on your recommendations, we have made revisions to the following sections. The specific changes are as follows (lines 377-396, lines 20-27):

1) Firstly, we have revised the Discussion section of the article (lines 377-396) as follows:

In this study, exogenous melatonin significantly promoted phenylalanine accumulation—a key precursor in anthocyanin biosynthesis—in the ZJ cultivar under lead (Pb) stress. Along with qPCR results and increased anthocyanin and catechin levels, this suggested melatonin might enhance phenylalanine conversion into flavonoids, consistent with previous findings on its role in regulating flavonoid biosynthesis (Hoque et al. 2021), In contrast, melatonin reduced phenylalanine levels in LJ43, while promoted its conversion mainly toward catechin synthesis.

Under non-stress conditions (MT vs. CK), despite significantly elevated endogenous melatonin, anthocyanin and catechin levels remained stable. This might reflect sufficient baseline flavonoid levels under normal physiology and the high energy cost of flavonoid biosynthesis, typically induced by stress (Agati et al. 2012). Melatonin also conferred stronger Pb tolerance in ZJ, likely due to its higher basal anthocyanin content and greater responsiveness in flavonoid synthesis, enhancing antioxidant capacity (Landi et al. 2015). However, current data were insufficient to determine whether melatonin-mediated Pb tolerance depends more on anthocyanin or catechin accumulation. Both are effective in scavenging ROS and chelating metal ions (Yu et al. 2024), but differ in localization, accumulation patterns, and physiological roles. Further studies are needed to identify which plays the dominant role in melatonin-induced heavy metal tolerance.

2) Secondly, we have revised lines 20-27 of the abstract as follows:

Moreover, exogenous melatonin significantly reduced Pb concentrations in roots, stems, and leaves, with a more pronounced effect in ZJ (reductions of 20.46%, 53.30%, and 38.17%, respectively), which might be associated with the downregulation of Pb transport genes like CsZIP1 (notably showing a 29-fold decrease). While these results suggested that melatonin might enhance Pb stress tolerance by modulating flavonoid metabolism and restricting Pb uptake, the specific roles of anthocyanins and catechins in this process remained to be fully elucidated. Further studies are necessary to clarify the primary bioactive compounds and mechanisms involved in melatonin-mediated heavy metal stress mitigation.

Comments 3: To better demonstrate the role of melatonin in regulating anthocyanin biosynthesis under Pb stress, a melatonin synthesis inhibitor such as pchlorophenylalanine is suggested to clarify the endogenous melatonin level in tea plants during Pb stress.

Response 3: Thank you very much for your valuable suggestion! Indeed, distinguishing the roles of exogenous and endogenous melatonin is important for elucidating the mechanisms underlying melatonin-mediated responses to Pb stress. However, previous studies have shown that melatonin, as a small lipophilic molecule, can readily permeate biological membranes (Salehi et al. 2019; Yu et al. 2016). Correspondingly exogenously applied melatonin may be able to enter plant cells directly and exert regulatory functions. Exogenous melatonin can directly penetrate cells to increase the endogenous melatonin levels in plants, which does not rely on the melatonin biosynthesis (Costa et al. 1995; Debnath et al. 2019). Therefore, this study focuses on the effects of exogenous melatonin on Pb stress in tea plants. Here, we are not concerned with the pathways through which exogenous melatonin increases the endogenous melatonin levels. In future, we will fully take your suggestions into consideration and incorporate melatonin inhibitor treatments to explore the pathways through which exogenous melatonin increases endogenous melatonin levels. This will be a new and promising research topic.

Minor Issues

Comments 1: As the latter half of this study focuses on anthocyanins and catechins, relevant background information should be briefly provided in the Introduction section.

Response 1: Thank you very much for your suggestion. We have added relevant background information on anthocyanins and catechins in the introduction section as follows (lines 88-100):

Melatonin can enhance anthocyanin concentration in plants by inducing the transcription of most anthocyanin biosynthesis genes and anthocyanin-related transcription factors (Liang et al. 2018; Sun et al. 2021). A similar effect has also been observed for catechins. Li et al. found that melatonin increased the transcription levels of catechin biosynthesis genes in tea leaves under moderately high-temperature stress (Li et al. 2020). Anthocyanins and catechins are both synthesized through the flavonoid biosynthetic pathway. Catechins are a major component of tea leaves, accounting for more than 75% of the tea polyphenols (Bae et al. 2020). They are primarily concentrated in chloroplasts and cell walls, where they exhibit powerful antioxidant properties. Catechins act as scavengers of reactive oxygen species and metal ion chelators, and they play a role in regulating photosynthesis. Additionally, they help protect plants from pathogens and environmental stress (Bernatoniene and Kopustinskiene 2018; Yu et al. 2024). Anthocyanins, which are widely distributed in various plant tissues, also function as antioxidants. They participate in both biotic and abiotic stress responses and can act as metal chelators (Stintzing and Carle 2004).

Comments 2: In the last paragraph of the Introduction, please briefly summarize the main findings rather than only stating the research aim.

Response 2: Based on your suggestion, we have added a summary of the results at the end of the introduction section as follows (lines 108-114):

This study aims to explore the mechanism by which exogenous melatonin enhances Pb tolerance in tea plants. Using pharmacological approaches and two divergent tea cultivars, we found that the tolerance of tea plants to Pb largely depends on the basal levels of anthocyanin. Moreover, we found that melatonin might enhance Pb stress tolerance by modulating flavonoid metabolism and restricting Pb uptake. This study provides new theoretical and practical guidance for enhancing tea plants' tolerance to heavy metal pollution.

Comments 3: Lines 139–143, please add figure numbers when describing the results.

Response 3: Thank you for your suggestion. We have added the figure number "Figure 2A" (now line 157) when describing the results.

Comments 4: Figures 2D–E are missing genotype labels (LJ43 and ZJ).

Response 4: According to your suggestion, we have added genotype labels (LJ43 and ZJ) in Figure 2D-E (line 184). In addition, we also found that genotype labels were missing in Figure 4A-F (line 235), and we have made the necessary revisions.

Comments 5: Lines 193–195, it is stated that polyphenols in ZJ under Pb stress decreased by 5.22% compared to CK. However, Figure 3B shows an increase. Please revise and provide clarification.

Response 5: Thank you very much for your feedback! We have carefully reviewed the data. The increase in polyphenol concentration in Figure 3B, line 214, is correct. The error lies in our description, which has now been revised in lines 208-209 to: "In contrast, in ZJ, polyphenol concentration increased by 5.22%, while flavonoid concentration decreased by 22.33%"

Comments 6: Figure 5 legend, there are no statistical comparisons in this figure, as stated in the figure legend. Please add relevant significance markers.

Response 6: According to your suggestion, we have added significance markers to the top right corner of the expression levels of each gene in Figure 5 (line 294).

Comments 7: In section 4.2 (Measurement Methods), part 1 on Pb content: Only leaf extraction is described, but Figure 2 also includes data on stem and root Pb concentration. Please add the extraction details for these tissues.

Response 7: Thank you very much for your suggestion! The methods for determining lead concentration in the roots, stems, and leaves of tea plants are the same in this study. Based on your advice, we have revised the description of the tea tree lead concentration extraction method in lines 487-488 from "Tea leaves were freeze-dried, ground" to "The tea tree leaves, stems, and roots were freeze-dried, ground, and 0.05 g of the sample was weighed into digestion tubes (Retka et al. 2010)."

References:

  1. Agati, G; Azzarello, E; Pollastri, S; Tattini, M. Flavonoids as antioxidants in plants: Location and functional significance. Plant Science 2012, 196, 67-76.
  2. Arnao, M B; Hernández‐Ruiz, J. Functions of melatonin in plants: a review. Journal of pineal research 2015, 59, 133-150.
  3. Bae, J; Kim, N; Shin, Y; Kim, S-Y; Kim, Y-J. Activity of catechins and their applications. Biomedical Dermatology 2020, 4, 1-10.
  4. Bernatoniene, J; Kopustinskiene, D M. The role of catechins in cellular responses to oxidative stress. Molecules 2018, 23, 965.
  5. Buttar, Z A; Wu, S N; Arnao, M B; Wang, C; Ullah, I; Wang, C. Melatonin Suppressed the Heat Stress-Induced Damage in Wheat Seedlings by Modulating the Antioxidant Machinery. Plants 2020, 9, 809.
  6. Costa, E J; Lopes, R H; Lamy‐Freund, M T. Permeability of pure lipid bilayers to melatonin. Journal of pineal research 1995, 19, 123-126.
  7. Debnath, B; Islam, W; Li, M; Sun, Y; Lu, X; Mitra, S; Hussain, M; Liu, S; Qiu, D. Melatonin Mediates Enhancement of Stress Tolerance in Plants. Int J Mol Sci 2019, 20.
  8. Ding, F; Liu, B; Zhang, S. Exogenous melatonin ameliorates cold-induced damage in tomato plants. Scientia Horticulturae 2017, 219, 264-271.
  9. Gamalero, E; Glick, B R. How Melatonin Affects Plant Growth and the Associated Microbiota. Biology 2025, 14, 371.
  10. Han, Q-H; Huang, B; Ding, C-B; Zhang, Z-W; Chen, Y-E; Hu, C; Zhou, L-J; Huang, Y; Liao, J-Q; Yuan, S. Effects of melatonin on anti-oxidative systems and photosystem II in cold-stressed rice seedlings. Frontiers in Plant Science 2017, 8, 785.
  11. Hoque, M N; Tahjib-Ul-Arif, M; Hannan, A; Sultana, N; Akhter, S; Hasanuzzaman, M; Akter, F; Hossain, M S; Sayed, M A; Hasan, M T. Melatonin modulates plant tolerance to heavy metal stress: morphological responses to molecular mechanisms. International journal of molecular sciences 2021, 22, 11445.
  12. Landi, M; Tattini, M; Gould, K S. Multiple functional roles of anthocyanins in plant-environment interactions. Environmental and experimental botany 2015, 119, 4-17.
  13. Li, N; Wang, J; Song, W-Y. Arsenic uptake and translocation in plants. Plant and Cell Physiology 2016, 57, 4-13.
  14. Li, X; Ahammed, G J; Zhang, X-N; Zhang, L; Yan, P; Zhang, L-P; Fu, J-Y; Han, W-Y. Melatonin-mediated regulation of anthocyanin biosynthesis and antioxidant defense confer tolerance to arsenic stress in Camellia sinensis L. Journal of Hazardous Materials 2021, 403, 123922.
  15. Li, X; Li, M-H; Deng, W-W; Ahammed, G J; Wei, J-P; Yan, P; Zhang, L-P; Fu, J-Y; Han, W-Y. Exogenous melatonin improves tea quality under moderate high temperatures by increasing epigallocatechin-3-gallate and theanine biosynthesis in Camellia sinensis L. Journal of plant physiology 2020, 253, 153273.
  16. Liang, D; Shen, Y; Ni, Z; Wang, Q; Lei, Z; Xu, N; Deng, Q; Lin, L; Wang, J; Lv, X. Exogenous melatonin application delays senescence of kiwifruit leaves by regulating the antioxidant capacity and biosynthesis of flavonoids. Frontiers in plant science 2018, 9, 426.
  17. Madigan, A P; Egidi, E; Bedon, F; Franks, A E; Plummer, K M. Bacterial and fungal communities are differentially modified by melatonin in agricultural soils under abiotic stress. Frontiers in Microbiology 2019, 10, 2616.
  18. Muhammad, I; Ullah, F; Ahmad, S; AlMunqedhi, B M; Al Farraj, D A; Elshikh, M S; Shen, W. A meta-analysis of photosynthetic efficiency and stress mitigation by melatonin in enhancing wheat tolerance. BMC Plant Biology 2024, 24, 427.
  19. Retka, J; Maksymowicz, A; Karmasz, D. Determination of Cu, Ni, Zn, Pb, Cd by ICP-MS and Hg by AAS in plant samples. 2010.
  20. Salehi, B; Sharopov, F; Fokou, P V T; Kobylinska, A; Jonge, L d; Tadio, K; Sharifi-Rad, J; Posmyk, M M; Martorell, M; Martins, N. Melatonin in medicinal and food plants: occurrence, bioavailability, and health potential for humans. Cells 2019, 8, 681.
  21. Sharma, P; Dubey, R S. Lead toxicity in plants. Brazilian journal of plant physiology 2005, 17, 35-52.
  22. Stintzing, F C; Carle, R. Functional properties of anthocyanins and betalains in plants, food, and in human nutrition. Trends in food science & technology 2004, 15, 19-38.
  23. Sun, H-l; Wang, X-y; Shang, Y; Wang, X-q; Du, G-d; LÜ, D-g. Preharvest application of melatonin induces anthocyanin accumulation and related gene upregulation in red pear (Pyrus ussuriensis). Journal of Integrative Agriculture 2021, 20, 2126-2137.
  24. Yu, H; Dickson, E; Jung, S-R; Koh, D-S; Hille, B. High membrane permeability for melatonin. Biophysical Journal 2016, 110, 605a.
  25. Yu, J; Wang, Q; Wang, W; Ma, R; Ding, C; Wei, K; Wang, L; Ge, S; Shi, Y; Li, X. Transcriptomic and metabolomic insights into temperature-dependent changes in catechin and anthocyanin accumulation in tea plants with different leaf colors. Plant Stress 2024, 14, 100705.
  26. Zhao, F J; Ma, J F; Meharg, A; McGrath, S. Arsenic uptake and metabolism in plants. New Phytologist 2009, 181, 777-794.

Round 2

Reviewer 2 Report

Comments and Suggestions for Authors

The manuscript has been improved after revision. Most of the issues pointed out in the previous review have been addressed, and the descriptions of the results and discussion have been revised to be more accurate, with reasonable explanations provided.

However, there are several formatting problems in the current version, particularly in the Results section, which make the text difficult to read and follow. For example, lines 264–272 are a repetition of lines 196–204, which discuss proline and soluble protein content, and should not appear again in Section 2.4. These issues must be carefully addressed before acceptance.

In addition, the reference issues highlighted in the previous review remain unresolved. For clarity, I reiterate them here:

  1. Lines 85–87, the sentence refers to the effects of melatonin in naked oat during Pb stress, but the cited references are on maize [36] and Ardisia species [37]. Please correct this.
  2. Lines 194–196, please add a reference to support the statement.
  3. Lines 205–206, please add a reference to support the statement.
Comments on the Quality of English Language

There are several spelling errors in the main text and figures. For example, in Figure 7, "Harnful effect" should be corrected to "Harmful effect." Please carefully check and revise all spelling and formatting issues throughout the manuscript.

Author Response

Comments and Suggestions for Authors

The manuscript has been improved after revision. Most of the issues pointed out in the previous review have been addressed, and the descriptions of the results and discussion have been revised to be more accurate, with reasonable explanations provided. However, there are several formatting problems in the current version, particularly in the Results section, which make the text difficult to read and follow.

Comments 1: However, there are several formatting problems in the current version, particularly in the Results section, which make the text difficult to read and follow. For example, lines 264–272 are a repetition of lines 196–204, which discuss proline and soluble protein content, and should not appear again in Section 2.4. These issues must be carefully addressed before acceptance.

Response 1: Thank you very much for your valuable suggestion. We have deleted the content from lines 264–272 and retained the section on proline and soluble proteins in the current manuscript at lines 196–204.

Comments 2: Lines 85–87, the sentence refers to the effects of melatonin in naked oat during Pb stress, but the cited references are on maize [36] and Ardisia species [37]. Please correct this.

Response 2: Thank you very much for your suggestion! We have revised the content to the current lines 85–87 as follows:

Additionally, melatonin effectively alleviated Pb stress in naked oat [37] by inducing DNA demethylation of metal transporter and antioxidant genes, and also mitigated the toxic effects of lead on maize [38] and Ardisia species [39].

References:

  1. Wang, K.; He, J.; Zhao, N.; Zhao, Y.; Qi, F.; Fan, F.; Wang, Y. Effects of melatonin on growth and antioxidant capacity of naked oat (Avena nuda L) seedlings under lead stress. PeerJ 2022, 10, e13978.
  2. Ullah, F.; Saqib, S.; Zaman, W.; Khan, W.; Zhao, L.; Khan, A.; Khan, W.; Xiong, Y.-C. Mitigating drought and heavy metal stress in maize using melatonin and sodium nitroprusside. Plant Soil 2024, 1-23.
  3. Ai, J.; Song, J.; Yan, Z.; Wang, Z.; Chen, W.; Wu, Y.; Wang, Y.; Pan, L.; Xu, Y.; Liu, P. Effects of Exogenous Melatonin on physiological response and DNA damage of Ardisia mamillata and A. crenata under lead stress. Chinese Bulletin of Botany 2022, 57, 171-181.

Comments 3: Lines 194–196, please add a reference to support the statement.

Response 3: Thank you for your suggestion. We have added two references to the content on lines 194-196, as follows:

Osmoregulation substances play a crucial role in helping plants cope with various stresses, such as drought, salt stress, and heavy metal stress, by maintaining cellular osmotic balance [49,50].

References:

  1. Hossain, A.; Ahmad, Z.; Adeel, M.; Rahman, M.A.; Alam, M.J.; Ahmed, S.; Aftab, T. Emerging roles of osmoprotectants in heavy metal stress tolerance in plants. In Heavy Metal Toxicity in Plants. CRC Press, 2021: 95-110.
  2. Zulfiqar, F.; Akram, N.A.; Ashraf, M. Osmoprotection in plants under abiotic stresses: New insights into a classical phenomenon. Planta 2020, 251, 3.

Comments 4: Lines 205–206, please add a reference to support the statement.

Response 4: Thank you sincerely for your valuable suggestion! We have added two references to the content on lines 205-206, as follows:

In addition, plant secondary metabolites play a crucial role in responding to heavy metal stress [51,52].

References:

  1. Michalak, A. Phenolic compounds and their antioxidant activity in plants growing under heavy metal stress. Polish Journal of Environmental Studies. 2006, 15(4).
  2. Upadhyay, R.; Saini, R.; Shukla, P.; Tiwari, K. Role of secondary metabolites in plant defense mechanisms: A molecular and biotechnological insights. Phytochemistry Reviews. 2025, 24, 953-983.

Comments 5: There are several spelling errors in the main text and figures. For example, in Figure 7, "Harnful effect" should be corrected to "Harmful effect." Please carefully check and revise all spelling and formatting issues throughout the manuscript.

Response 5: Thank you very much for your feedback! We have carefully checked the spelling and formatting issues in the manuscript and made the following revisions (lines 428, 436-437, 555, 591):

(1) We have corrected "Harnful effect" to "Harmful effect" in Figure 7 at line 428.

(2) We have changed "Decreased" to "Decrease," "Increased" to "Increase," "Boosts" to "Boost," "Reduces" to "Reduce", and "Promotes" to "Promote" in Figure 7.

(3) We have changed " he roots of LJ 43 and ZJ were thoroughly washed and cultured in water for one week." to " the roots of LJ 43 and ZJ were thoroughly washed and grown in water for one week." in line 436-437.

(4) We modified the "P" on line 555 to italics.

(5) We updated the abbreviation list on line 591 by capitalizing the first letter of the full terms.
